# Centriolar remodeling underlies basal body maturation during ciliogenesis in *Caenorhabditis elegans*

**Inna V Nechipurenko[1][*][†], Cristina Berciu[2][†][‡], Piali Sengupta[1][*], Daniela Nicastro[2,3][*]**

[1]Department of Biology and National Center for Behavioral Genomics, Brandeis University, Waltham, United States; [2]Department of Biology and Rosenstiel Basic Medical Sciences Research Center, Brandeis University, Waltham, United States; [3]Departments of Cell Biology and Biophysics, University of Texas Southwestern Medical Center, Dallas, United States

**Abstract** The primary cilium is nucleated by the mother centriole-derived basal body (BB) via as yet poorly characterized mechanisms. BBs have been reported to degenerate following ciliogenesis in the *C. elegans* embryo, although neither BB architecture nor early ciliogenesis steps have been described in this organism. In a previous study (Doroquez et al., 2014), we described the three-dimensional morphologies of sensory neuron cilia in adult *C. elegans* hermaphrodites at high resolution. Here, we use serial section electron microscopy and tomography of staged *C. elegans* embryos to demonstrate that BBs remodel to support ciliogenesis in a subset of sensory neurons. We show that centriolar singlet microtubules are converted into BB doublets which subsequently grow asynchronously to template the ciliary axoneme, visualize degeneration of the centriole core, and define the developmental stage at which the transition zone is established. Our work provides a framework for future investigations into the mechanisms underlying BB remodeling.

**\*For correspondence:** ivn@ brandeis.edu (IVN); sengupta@ brandeis.edu (PS); daniela. nicastro@utsouthwestern.edu (DN)

[†]These authors contributed equally to this work

**Present address:** [‡]Microscopy Core Facility McLean Hospital, Belmont, United States

## Introduction

Cilia are evolutionarily conserved microtubule (MT)-based organelles that play key roles in regulating embryonic development, sensory signaling, and motility among other cellular functions (*Goetz and Anderson, 2010*; *Green and Mykytyn, 2010*; *Yildiz and Khanna, 2012*; *Falk et al., 2015*). Both immotile primary and motile cilia are nucleated by a basal body (BB) that is generally derived from the mother centriole (*Marshall, 2007*; *Kim and Dynlacht, 2013*). BBs possess accessory structures such as transition fibers that associate with a ciliary vesicle or dock with the plasma membrane and provide a platform for assembly of intraflagellar transport (IFT) complexes that are essential for elongation of the ciliary axoneme (*Kim and Dynlacht, 2013*; *Azimzadeh and Marshall, 2010*; *Reiter et al., 2012*; *Dawe et al., 2007*). Although overall organization of centrioles/BBs, as well as many proteins required for their assembly and function are conserved, ultrastructural features of these cellular structures can differ among and within species. For instance, centrioles/BBs are cylindrical structures that can be comprised of a radially symmetric array of MT singlets, doublets, or triplets depending on the species and cellular context (*Azimzadeh and Marshall, 2010*; *Winey and O'Toole, 2014*; *Carvalho-Santos et al., 2011*; *Gottardo et al., 2015*; *González et al., 1998*; *Jana et al., 2016*). It remains unclear whether centrioles of distinct ultrastructural organization transition to BBs and nucleate cilia via similar or distinct mechanisms.

Ultrastructural analyses of centrioles in one-cell *C. elegans* embryos have shown that centrioles in this organism are structurally distinct from their mammalian counterparts (*Pelletier et al., 2006*). *C. elegans* centrioles are relatively small compared to those in mammals and are comprised of a central

tube surrounded by nine singlet MTs (sMTs), as compared to the cartwheel structure surrounded by triplet MTs found in larger mammalian centrioles (*Winey and O'Toole, 2014*; *Pelletier et al., 2006*; *Hilbert et al., 2013*; *Gönczy, 2012*). Despite these differences, *C. elegans* and vertebrate centrioles are built using subsets of conserved proteins (*Carvalho-Santos et al., 2011*; *Gönczy, 2012*).

Primary cilia are present only on sensory neurons in *C. elegans* (*Ward et al., 1975*; *Perkins et al., 1986*). As in other organisms, these cilia are templated by BBs derived from centrioles (*Perkins et al., 1986*). However, BBs in *C. elegans* have been reported to degenerate following cilia assembly in the embryo, and no canonical BB structures or core centriolar components are detected in ciliated neurons in wild-type animals at postembryonic stages (*Perkins et al., 1986*; *Dammermann et al., 2009*; *Schouteden et al., 2015*; *Doroquez et al., 2014*). Intriguingly, despite this apparent degeneration, a subset of BB-associated proteins remains enriched at the cilia base in adult animals as shown via immunofluorescence (*Dammermann et al., 2009*; *Mohan et al., 2013*; *Wei et al., 2013*, *2016*), suggesting the presence of centriolar/BB 'remnants'. Since sensory neurons are born and differentiate at late embryonic stages (*Sulston et al., 1983*) that are technically challenging to analyze experimentally, key early steps in ciliogenesis including the centriole-to-BB transition, the precise timing of centriolar degeneration, and initiation of axoneme elongation, have yet to be examined in this organism.

In a recent report, we described the three-dimensional morphologies of sensory cilia in the nose of *C. elegans* hermaphrodites at high resolution using serial section transmission electron microscopy (ssTEM) and serial section electron tomography (ssET) of high pressure-frozen and freeze-substituted (HPF-FS) adult animals (*Doroquez et al., 2014*). Here, we use these imaging methods to describe early steps of ciliogenesis in the *C. elegans* embryo. We find that sMTs of centrioles in early embryos contain hook-like appendages that remodel to dMTs during BB maturation and prior to axoneme elongation, and template the dMTs of the ciliary axoneme. We show that these BB dMTs at the cilia base 'flare' at later embryonic stages, and that this flaring coincides with degeneration of the central tube of the centriole/BB. We also visualize formation of the transition zone (TZ), a compartment that acts as a diffusion barrier at the ciliary base, and the apical ring, a structure present at the distal TZ (*Perkins et al., 1986*; *Doroquez et al., 2014*; *Blacque and Sanders, 2014*). Our observations indicate that the centriole/BB does not fully degenerate, but that the outer centriole wall remodels to nucleate the axoneme and persists through adulthood in a subset of *C. elegans* sensory neurons. This work reports key early steps in BB maturation and ciliogenesis and extends our previous ultrastructural analyses of adult sensory cilia in this organism.

## Results

### Hook-like appendages of A-tubules close to form the B-tubules of the BB and axoneme in a subset of *C. elegans* ciliated sensory neurons

Twelve pairs of ciliated sensory neurons are found in the bilateral amphid sensory organs of the head in the *C. elegans* hermaphrodite (*Ward et al., 1975*; *Perkins et al., 1986*). Eight of these neurons extend their simple rod-like cilia through a channel created by glial cells (*Ward et al., 1975*; *Perkins et al., 1986*; *Doroquez et al., 2014*) (*Figure 1—figure supplement 1*). Since these channels, and neuronal endings contained therein, are readily identifiable in serial sections of the embryo, we focused our attention on this subset of ciliated cells. Amphid sensory neurons are born over a period of time from the end of ventral closure to the comma stage of embryogenesis (*Sulston et al., 1983*) (*Figure 1A*), and cilia of 3 μm or longer have been previously detected in these neurons starting at the three-fold stage using fluorescent reporters (*Fujiwara et al., 1999*). However, core centriole proteins such as SAS-4 are not detected at these later developmental times (*Dammermann et al., 2009*; *Schouteden et al., 2015*; *Kirkham et al., 2003*) (*Figure 1A*), suggesting that the centriole has at least partly degenerated, and that cilia have elongated by these late embryonic stages.

To visualize centrioles and cilia in the embryo, we examined wild-type *C. elegans* embryos at multiple developmental stages using ssTEM and ssET. Although a subset of amphid neurons has already been born by 350 min post-fertilization (mpf) (*Figure 1A*), we were unable to detect amphid channels formed by glial cells in embryos of this stage (*Figure 1—figure supplement 2A*), and thus, could not unambiguously distinguish amphid sensory neurons from other cell types. Nevertheless,

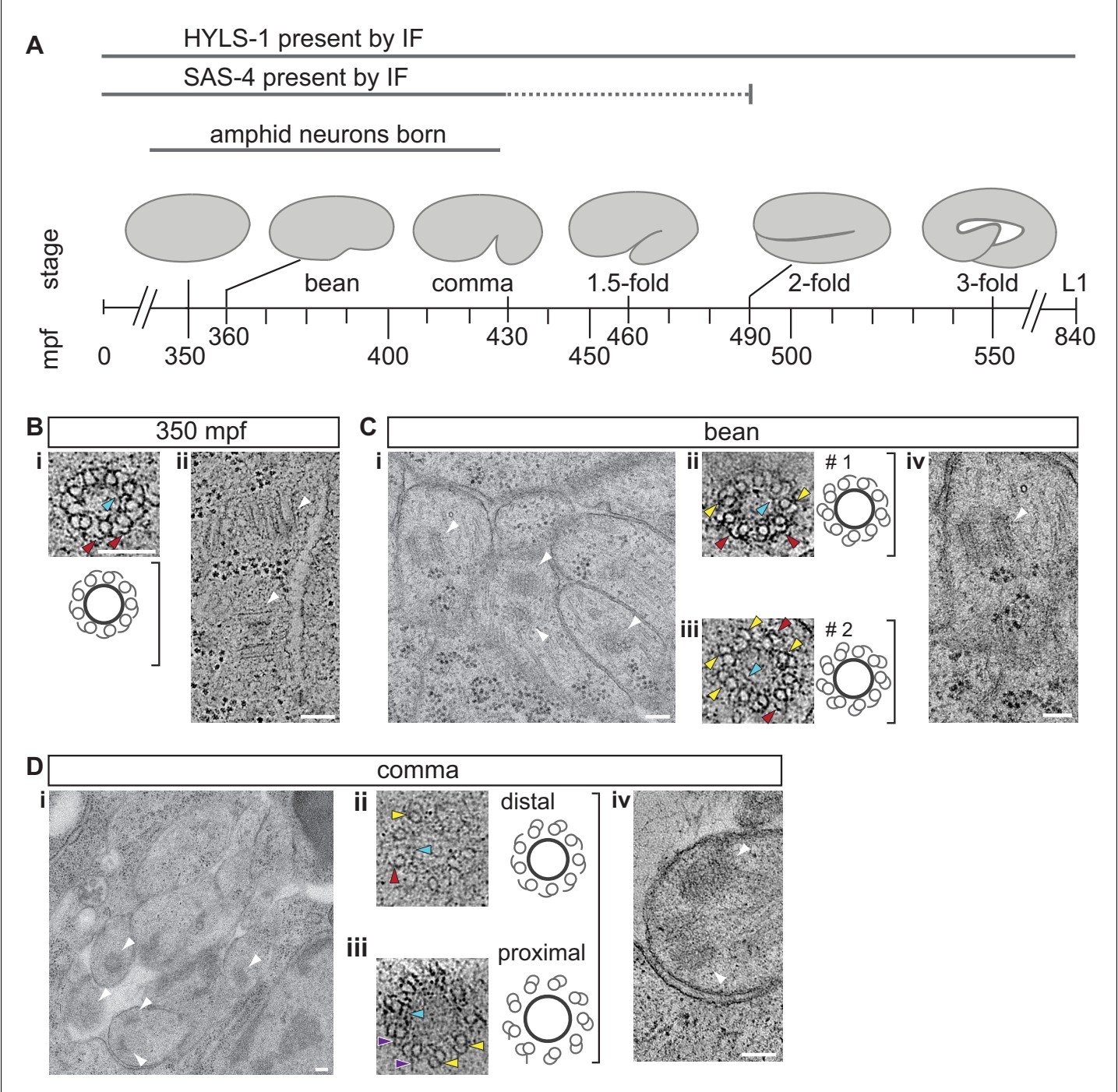

**Figure 1.** The centrioles of embryonic amphid sensory neurons remodel to initiate ciliogenesis. (**A**) Timeline of embryonic development at 22°C (adapted from IntroFIG7 http://www.wormatlas.org/ver1/handbook/anatomyintro/anatomyintro.htm) showing select stages defined by morphology between fertilization (t = 0) and hatching (L1). The approximate time period during which amphid neurons are born is marked. The developmental stages during which core centriole components (e.g. SAS-4 and BB-associated protein HYLS-1 [*Dammermann et al., 2009*; *Wei et al., 2016*]) are detected by immunofluorescence (IF) in amphid neurons are marked. Dashed line indicates that the exact time of SAS-4 loss in amphid neurons is unknown. mpf – minutes post fertilization. (**B–D**) TEM images of the amphid channel in cross-section (**Ci** and **Di**), cross-section ET slices and schematics of a centriole in an unidentified cell of a 350 mpf embryo (**Bi**) and BBs in amphid neurons (**Cii, Ciii, Dii, Diii**), and ET slices showing centrioles/BBs in longitudinal orientation (**Bii, Civ, Div**) at the indicated stages of embryogenesis. Two different examples of bean-stage centrioles (#s 1 and 2) undergoing remodeling are shown in **Cii** and **Ciii**, respectively. Arrowheads: centrioles/BBs (white), dMTs (yellow), sMTs with hook appendages (red), central tube (light blue), putative nascent Y-links (purple). Scale bars: 100 nm.

*Figure 1 continued on next page*

*Figure 1 continued*

The following figure supplements are available for figure 1:

**Figure supplement 1.** The cilia of a subset of amphid sensory neurons extend through a channel created by glial support cells.

**Figure supplement 2.** Example TEM cross-section images of *C. elegans* embryos at the 350 mpf and bean stages.

consistent with previous reports (*Pelletier et al., 2006*; *Delattre et al., 2004*; *Mikeladze-Dvali et al., 2012*), we observed centrioles comprised of the central tube surrounded by sMTs with hook-like appendages in many cells (*Figure 1B*). The average diameter and length of centrioles at this developmental stage were $88.2 \pm 4.2$ nm and $100.7 \pm 0.2$ nm, respectively (Figure 3A).

We next examined serial sections of embryos frozen at the 'bean' stage (*Figure 1A*). In contrast to our observations at 350 mpf, bilateral amphid channels containing sensory neuron endings were readily visible at this and all subsequent stages (*Figure 1—figure supplement 2B*, *Figure 2—figure supplement 1*). Centrioles in this stage were not located in close proximity to the cell surface and dendritic tip but were instead found deep within the cell (*Figure 1Ci and iv*). Interestingly, at this stage, we observed a transition from sMTs with hooks to dMTs. Specifically, centrioles in several amphid neurons were comprised of the central tube surrounded by a mixture of dMTs and sMTs with hooks (*Figure 1Cii–iii*). The average length of these structures at the bean stage was similar to that at the 350 mpf stage (*Figure 3B*). We, therefore, infer that sMTs remodel to form dMTs by closure of the A tubule-associated hooks to generate B-tubules. As these centrioles contain both dMTs and sMTs with hooks within the same 70 nm section, the transition from sMTs to dMTs likely occurs asynchronously within a centriole. We henceforth refer to this remodeled structure as the BB, and conclude that this remodeling is initiated by the bean stage of embryonic development in a subset of amphid sensory neurons.

By the 'comma' stage (*Figure 1A*), BBs in a subset of amphid neurons were found in close proximity to the cell surface (*Figure 1Di and iv*). Although the average length of these structures was mildly increased, their mean diameter and length were not significantly different relative to those at the bean stage (*Figure 3A–B*). At the comma stage, dMTs surrounded the central tube in the most proximal BB regions; however, a mixture of dMTs and sMTs with hooks was present in more distal BB/axoneme regions in most examined cells (*Figure 1Dii–iii*). As BBs/axonemes elongate in later stage embryos (*Figure 2A–C*, *Figure 3B*), only dMTs were detected in proximal regions, whereas a mixture of dMTs and sMTs with hooks were present in more distal regions of axonemes in 1.5- and two-fold embryos (*Figure 2A–C*). These observations suggest that A- and B-tubules of BBs/axonemes grow asynchronously in examined amphid sensory neurons (summarized in *Figure 3C*).

## Degeneration of the centriole core coincides with increased ciliary base diameter

We previously reported flaring of dMTs at the ciliary base in adult amphid neurons, and proposed that this flaring is a consequence of BB degeneration (*Doroquez et al., 2014*). We found that in embryos, the diameter of the proximal BB/axoneme region was variable at the 1.5-fold stage and significantly increased by the two-fold stage (*Figure 3A*). Previous observations have reported loss of core centriolar/BB markers such as SAS-4 by the two-fold stage by immunofluorescence (*Dammermann et al., 2009*; *Schouteden et al., 2015*), raising the possibility that the increased BB diameter is a consequence of degeneration of core centriolar structures.

All examined cross-sections of centrioles/BBs in 350 mpf embryos as well as bean and comma-stage amphid neurons contained the central tube with an average diameter of $60.9 \pm 4.2$ nm (*Figure 1B–D*). However, the central tube appeared to be present in only a subset of examined BBs in 1.5-fold embryos (compare *Figure 2Aiv* with *Figure 2Av*, *Figure 2—figure supplement 2*) and was absent from all examined BBs in two- and three-fold embryos (*Figure 2B–C*). The diameter of the proximal regions of BBs with seemingly degenerated central tubes was larger than that of BBs with intact central tubes (compare *Figure 2Av* with *Figure 2Aiv*, *Figure 2Bv and Cv*). Consistent with a significant increase in proximal BB/axoneme diameter evident from cross-sections, we observed flaring of dMTs at the ciliary base in longitudinal sections in a subset of amphid neurons at the 1.5-fold stage (*Figure 2Ai and Avii*), and in all examined amphid neurons at the two- and three-

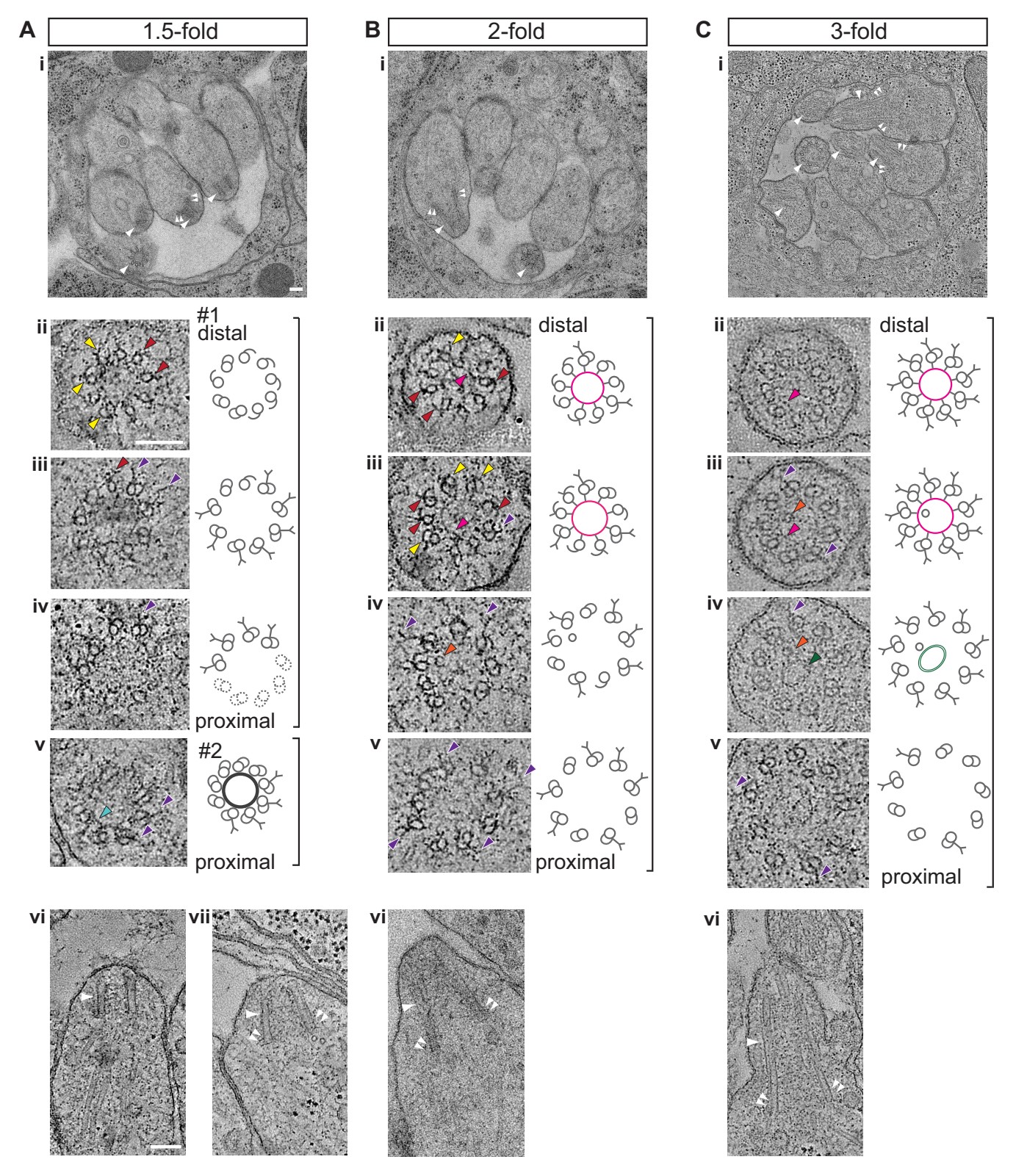

**Figure 2.** Degeneration of the central tube and dMT flaring at the ciliary base are observed by the 1.5-fold stage of embryogenesis. (A–C) TEM images (Ai and Bi) and ET slice (Ci) of the amphid channel in cross-section, cross-section ssET images and schematics of BBs in amphid neurons (Aii–v, Bii–v, Cii–v), and ET slices (Avi, Avii, Cvi) and TEM image (Bvi) showing BBs/axonemes in longitudinal orientation at the indicated stages of embryogenesis. Cii–v show a subset of ET slices from the serial section tomogram. Examples of BBs/axonemes in 1.5-fold embryo with a largely degenerated central

*Figure 2 continued on next page*

*Figure 2 continued*

tube and an intact central tube are shown in **Aiv** and **Av**, respectively. Arrowheads: centrioles/BBs (large white), flared dMTs at cilia base (small white double), dMTs (yellow), sMTs with hook appendages (red), central tube (light blue), Y-links (purple), apical ring (pink), isMTs (orange), a vesicle (green). Each bracket delineates a single BB/axoneme with its proximal and distal regions marked accordingly. Scale bars: 100 nm.

The following figure supplements are available for figure 2:

**Figure supplement 1.** Example TEM cross-section images of *C. elegans* embryos at the 1.5-, two-, and three-fold stages.

**Figure supplement 2.** Selected ET slices of tomograms of a 1.5-fold BB/axoneme reconstructed from sequential plastic sections.

**Figure supplement 3.** Selected ET slices of tomograms of two- and three-fold cilia showing isMTs.

fold embryonic stages (*Figure 2Bi, Bvi*, *3Ci and Cvi*). These results suggest that central tubes degenerate asynchronously in individual amphid neurons, and that degeneration of the central tube starting at the 1.5-fold stage likely accounts for the flaring of BB dMT arrays and increased BB diameters (*Figure 3A and C*). As we reported previously (*Doroquez et al., 2014*), we did not detect any obvious structures resembling transition fibers associated with BBs in *C. elegans* amphid neurons at any developmental stage by TEM of HPF-FS samples.

## The transition zone is formed by the 1.5-fold stage of embryogenesis in a subset of sensory neurons

The TZ at the cilia base is the proximal-most compartment of the axoneme proper. This compartment is defined ultrastructurally by the presence of proteinacious Y-links that originate at the outer junction between A- and B-tubules of axonemal dMTs, and project toward and usually connect to the plasma membrane (*Reiter et al., 2012*; *Blacque and Sanders, 2014*; *Czarnecki and Shah, 2012*). We investigated when Y-link structures are first observed ultrastructurally during axoneme elongation. Analyses of cross-sections identified obvious Y-shaped fibers emanating from dMTs at the 1.5-, two-, and three-fold stages (*Figure 2A–C*). At the comma stage, we observed shorter fiber-like densities without fully formed Y-link endings extending from subsets of dMTs (*Figure 1Diii*), potentially representing nascent Y-links. These observations suggest that structural features of TZs of a subset of amphid sensory neurons are at least partly established by the 1.5-fold stage of embryogenesis.

A puzzling feature of amphid sensory cilia is the presence of a variable number of inner singlet MTs (isMTs) inside adult axonemes (*Perkins et al., 1986*; *Doroquez et al., 2014*). These isMTs are smaller in diameter compared to A-tubules of the centriole/BB/axoneme and contain 11 protofilaments similar to cytoplasmic MTs in *C. elegans* (*Perkins et al., 1986*; *Doroquez et al., 2014*; *Chalfie and Thomson, 1979*). We defined isMTs as fully closed MTs that were observed inside axonemes over multiple tomographic slices. Using these criteria, we observed only one isMT in one axoneme at the two-fold stage and multiple isMTs in all examined axonemes at the three-fold stage (*Figure 2Biv, Ciii and Civ*; *Figure 2—figure supplement 3*). However, we detected incompletely closed MTs and short MT-like structures inside multiple axonemes at the two-fold stage (*Figure 2—figure supplement 3A*); these structures may represent early stages of isMT assembly. We also noted a ring-like structure in distal regions of the TZs in two- and three-fold axonemes (*Figure 2Bii–iii, Cii–iii*). This structure is likely the apical ring hypothesized to provide an attachment site for isMTs (*Perkins et al., 1986*; *Doroquez et al., 2014*; *Blacque and Sanders, 2014*). The origin and function of the isMTs remain to be determined.

## Discussion

Our observations suggest that in a subset of ciliated sensory neurons in the head amphid organs of *C. elegans*, the outer centriole wall is remodeled to initiate ciliogenesis and persists thereafter into adulthood, while the centriole core degenerates starting at the 1.5-fold embryonic stage (summarized in *Figure 3C*). This remodeling is consistent with the persistence of a subset of outer centriole wall- and centriole-associated proteins (eg. HYLS-1 and DYF-19/FBF1) through postembryonic stages

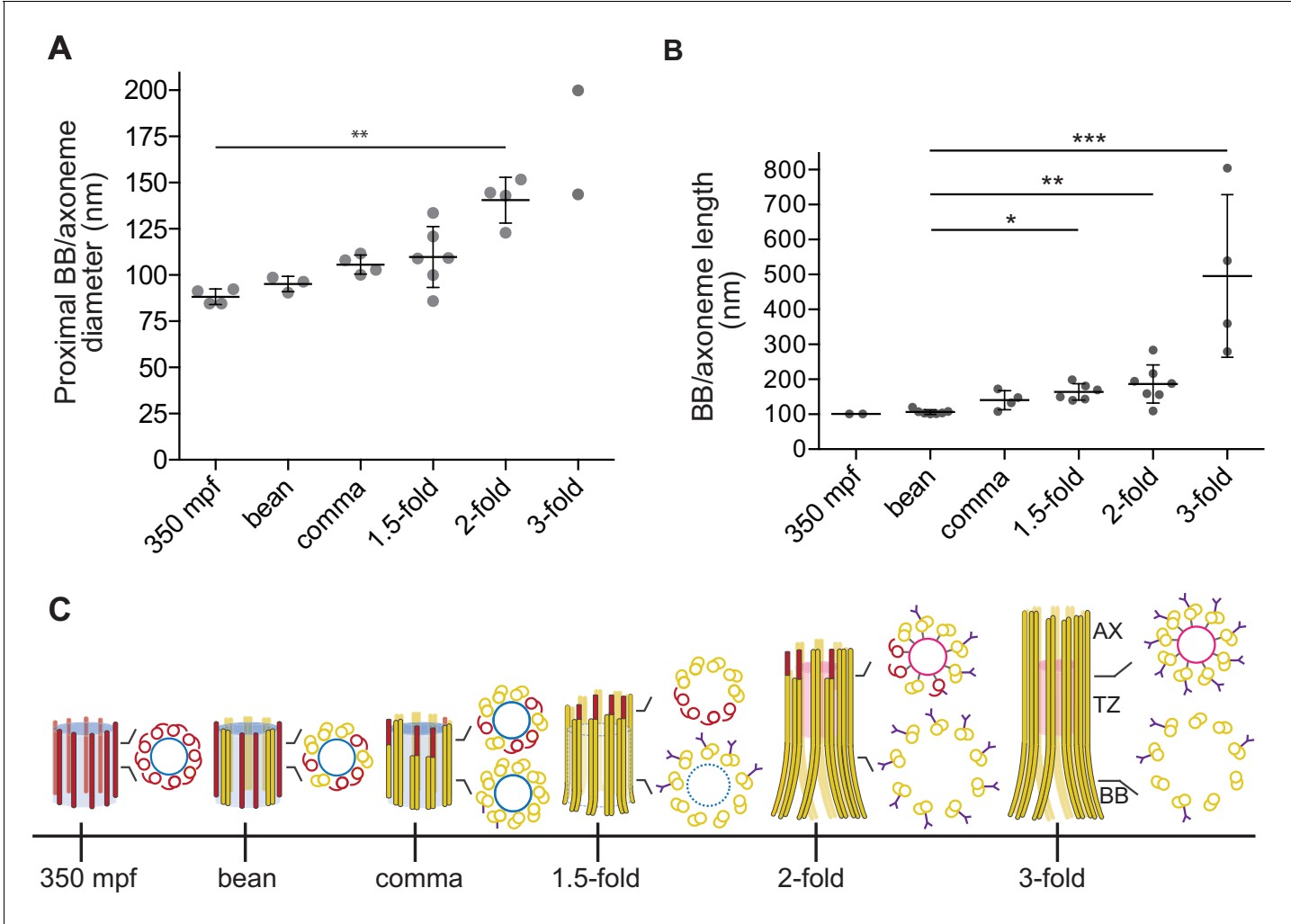

**Figure 3.** Quantification of BB/axoneme diameter and length in embryonic amphid sensory neurons. (**A**) Quantification of the centriole/BB/axoneme diameter measured as the distance between centers of A-tubules. The first tomographic slice of a ssET sequence showing the entire proximal region of each BB/axoneme in cross-section was used for measurements. Each dot represents a measurement from an individual BB/axoneme in different neurons from the same embryo. Horizontal bars indicate mean. Errors are SD. ** indicate that marked data sets are different at p<0.01 (Kruskal-Wallis test with post-hoc correction for multiple comparisons). (**B**) Quantification of the centriole/BB/axoneme length at the indicated stages of embryonic development. Each dot represents a measurement from an individual BB/axoneme in different neurons from the same embryo. Horizontal bars indicate mean. Errors are SD. *, **, and *** indicate that marked data sets are different at p<0.05, 0.01, and 0.001, respectively (Kruskal-Wallis test with post-hoc correction for multiple comparisons). (**C**) Model summarizing key early ciliogenesis stages in the examined subset of *C. elegans* sensory neurons. BB – basal body; TZ – transition zone; AX – axoneme. Blue and pink circles indicate the central tube and apical ring, respectively. Model is based solely on our ability to visualize specific ciliary structures, and no assumptions are made regarding the presence or absence of proteins associated with these structures.

(*Dammermann et al., 2009*; *Mohan et al., 2013*; *Wei et al., 2013*, *2016*). Thus, amphid channel cilia in adult *C. elegans* hermaphrodites do not completely lack a BB, but instead possess a modified BB at their base. Proteins (e.g. kinesin-II, DYF-19/FBF1, and HYLS-1) localized to this region in adult cilia have previously been implicated in loading, ciliary import, and recycling of IFT machinery (*Wei et al., 2013*, *2016*; *Prevo et al., 2015*), suggesting that this modified BB retains at least a subset of the functions of canonical BBs in other organisms.

The observed closure of sMT-associated hooks to form dMTs in *C. elegans* BBs is reminiscent of observations in *Drosophila*. Although mature centrioles generally contain nine dMTs in the *Drosophila* embryo, centrioles consisting of nine sMTs with lateral hooks have also been observed, and it has been speculated that these structures represent intermediates in the centriole assembly process

(*Gottardo et al., 2015*). Similar to flagella in *Chlamydomonas*, where B-tubules are assembled by tubulin addition onto A-tubules from the outer to inner AB junction (*Nicastro et al., 2011*; *Linck and Stephens, 2007*), B-tubules in *C. elegans* amphid BBs appear to form via closure of A-tubule-associated hooks at the inner AB junction. Intriguingly, B-tubules open at the inner AB junction have been observed in *C. elegans* and mice mutant for the TZ component NPHP-4 and the small GTPase Arl13b, respectively, raising the possibility that these defects may be a consequence of failure to close this junction during development (*Jauregui et al., 2008*; *Caspary et al., 2007*). Similar to our findings, asynchronous growth of dMTs comprising the outer centriole wall has been reported in *Paramecium* (*Dippell, 1968*), *Chlamydomonas* (*Preble et al., 2001*), human lymphoblasts (*Guichard et al., 2010*), and *Drosophila* germline stem cells (*Gottardo et al., 2015*). The mechanisms that mediate centriole wall remodeling are unknown.

Similar to *Drosophila*, BBs in *C. elegans* appear to lack structurally visible appendages (*Gottardo et al., 2015*; *González et al., 1998*; *Doroquez et al., 2014*; *Callaini and Riparbelli, 1990*; *Callaini et al., 1997*). As proposed previously, the absence of transition fibers suggests that BBs in *C. elegans* dock to the cell membrane via alternate mechanisms possibly requiring interaction of the BB and/or TZ with cell adhesion molecules (*Schouteden et al., 2015*; *Williams et al., 2011*; *Nechipurenko et al., 2016*; *Heiman and Shaham, 2009*). However, we note that in HPF-FS specimens, mesh-like structures are often less clearly visible than in conventional chemically fixed specimens in which fibrous networks can collapse into dark electron-dense appearing structures (*McEwen et al., 1998*). Thus, it remains possible that in *C. elegans*, transition fiber-associated proteins are organized in a less compact structure that is distinct from that in vertebrates. We detect structures resembling putative nascent Y-links of the TZ associated with dMTs in the comma stage when the central tube has not yet degenerated. At the 1.5-fold stage, clear Y-links are associated with BBs that either lack or contain the central tube, suggesting that while initiation of Y-link formation may require the centriole core, the core is dispensable for axoneme elongation. We find that loss of the central tube coincides with widening of the cilia base starting at the 1.5-fold stage of embryogenesis, suggesting that dMT flaring at the ciliary base is a consequence of central tube loss. We note that the timing of early ciliogenesis events may be distinct in ciliated cell types that were not examined in this study. In the future, it will be important to correlate the presence (or absence) of distinct subciliary structures with that of their known molecular components in individual cell types across developmental stages in order to obtain a more complete description of early ciliogenic steps. We expect that the ability to visualize centrioles/BBs and cilia in single cells in vivo together with the genetic power of *C. elegans* will allow further characterization of the conserved and species-specific mechanisms that underlie biogenesis and maintenance of these important cellular organelles.

## Materials and methods

### Strains
Wild-type strain of *C. elegans* (Bristol N2) was obtained from the *Caenorhabditis* Genetics Center and cultured on standard nematode growth media plates seeded with *E. coli* OP50.

### Specimen preparation
Gravid hermaphrodites were cut open to release eggs, which were then allowed to develop to the desired stage at room temperature in M9 buffer. Bean, comma, 1.5-fold, two-fold, and three-fold embryos were identified based on morphology. To collect 350 mpf embryos, single-cell embryos were manually sorted and allowed to develop for ~350 mins in M9 at 22°C. Embryos of the desired developmental stage were suctioned into cellulose capillary tubes (200 μm diameter, Leica Microsystems) in M9 and sealed.

The HPF-FS preparation was performed as described previously (*Doroquez et al., 2014*). Briefly, embryos in sealed capillary tubes were placed in the cavity between two aluminum planchettes (type 'A' hat, 100 μm deep, and the flat surface of type 'B' hat, Wohlwend, Switzerland) that was filled with 20% bovine serum albumin (BSA) in M9. The quickly assembled planchette sandwich was rapidly high pressure-frozen using a Leica EM HPM100 HPF system (Leica Microsystems, Vienna, Austria). Freeze-substitution was performed at −90°C over 3–4 days in fixation solution [1% osmium tetroxide

(19100, EMS), 0.5% glutaraldehyde (16530, EMS), 2% water in anhydrous acetone (AC32680-1000, Fisher)] using a Leica EM AFS2 FS system, before the temperature was progressively increased to 4°C (5°C/hr). After 1 hr at 4°C, samples were washed with anhydrous acetone (4 × 30 mins), infiltrated and flat embedded in Araldite 502/Embed-812 Resin [Araldite (10900, EMS), Embed-812 (14900, EMS), DDSA (13710, EMS)] at room temperature, and polymerized at 60°C for several days. Flat-embedded samples were subsequently re-embedded in order to obtain cross sections of embryos.

## Serial section TEM and ET

Serial sectioning, electron microscopy and tomography, and image processing were performed as described previously (*Doroquez et al., 2014*). Briefly, serial plastic sections (70-nm thick) were collected on slot grids covered with Formvar support film, post-stained with saturated solution of uranyl acetate (0379, Polysciences, Inc., Warrington, PA) for 15 min, and Reynold's lead citrate (Lead nitrate - 17900, EMS, and Sodium citrate - S-279, Fisher) for 7 min, and imaged using a Tecnai F20 (200 keV) or F30 (300 keV) transmission electron microscope (FEI, Hillsboro, OR) equipped with a 2K ×2K charged-coupled device (CCD) camera. For large overviews of sections, we acquired montages of overlapping high-magnification images. For electron tomography, BSA-coated, 10 nm colloidal gold fiducials (Au - Sigma-Aldrich, St. Louis, MO; BSA - SC-2323, Santa Cruz Biotechnology, Inc.) (*Iancu et al., 2006*) were applied to the sections, before acquiring dual-axis tilt series with a tilt range of ±60° with 1° increments around each axis. Automated montage and tilt series acquisition was facilitated by the microscope control software SerialEM (*Mastronarde, 2005*). Image processing, such as blending montages and reconstructing tomograms, was performed using various tools from the IMOD software package (*Kremer et al., 1996*).

# Acknowledgements

We are grateful to Alex Dammermann, Max Heiman and Shai Shaham for discussions and communication of data prior to publication. We thank Oliver Blacque for comments on the manuscript. We are grateful to Chen Xu for providing training and for management of the Brandeis EM facility. This work was supported in part by the NIH (R37 GM56223 – PS and R01 GM083122 – DN).

# Additional information

### Competing interests

PS: Reviewing editor, *eLife*. The other authors declare that no competing interests exist.

### Funding

| Funder | Grant reference number | Author |
| --- | --- | --- |
| National Institutes of Health | R37 GM56223 | Piali Sengupta |
| National Institutes of Health | R01 GM083122 | Daniela Nicastro |

The funders had no role in study design, data collection and interpretation, or the decision to submit the work for publication.

### Author contributions

IVN, Data curation, Formal analysis, Validation, Investigation, Visualization, Methodology, Writing—original draft, Writing—review and editing; CB, Formal analysis, Validation, Investigation, Visualization, Methodology; PS, Conceptualization, Supervision, Funding acquisition, Visualization, Writing—original draft, Project administration, Writing—review and editing; DN, Conceptualization, Formal analysis, Supervision, Funding acquisition, Visualization, Project administration, Writing—review and editing

Author ORCIDs

Inna V Nechipurenko, http://orcid.org/0000-0003-0249-6620

Piali Sengupta, http://orcid.org/0000-0001-7468-0035

Daniela Nicastro, http://orcid.org/0000-0002-0122-7173

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
