## [Decision Letter]

Thank you for submitting your article "Centriolar remodeling underlies basal body maturation during ciliogenesis in *Caenorhabditis elegans*” for consideration by *eLife*. Your article has been reviewed by two peer reviewers, and the evaluation has been overseen by Oliver Hobert as Reviewing Editor and K VijayRaghavan as the Senior Editor.

The reviewers have discussed the reviews with one another and the Reviewing Editor has drafted this decision to help you prepare a revised submission.

The manuscript by Nechipurenko et al. is a logical follow up from authors' previous *eLife* paper defining an ultrastructural map of *C. elegans* sensory cilia and glia. Here, the authors again use ssTEM and ssET to capture early steps of ciliogenesis in rapidly forming cilia in *C. elegans* embryos, which is a difficult and heroic accomplishment in microscopy. The authors make a number of very interesting and unexpected findings that are worthy of publication pending some revisions.

1) Many of the inferred temporal events presented in the concluding model appear somewhat speculative in the absence of some type of complementary real-time visualization using light microscopy. The authors need to better emphasize the caveats associated with their deductions.

2) Some of the implications of the work seem somewhat overstated – e.g. in the Discussion section the authors conclude that "[…]cilia in adult *C. elegans* […] possess a modified BB[…]that[…]provide(s) a site for docking, assembly and turnaround of IFT proteins…" but what is the evidence that these 3 processes require the modified BB? the authors should consider discussing their speculations about the import of the IFT machinery in the context of the paper by Prevo et al., 2015, Nat Cell Biol, 17, 1536 which reports the specific role of heterotrimeric kinesin-2 in importing the IFT machinery through structures described in the current paper at the base of the cilium.

3) Some implications of the authors findings are not emphasized in this manuscript and the authors are encouraged to discuss implications of their findings that will appeal to a broader audience. Overstatements should be avoided and caveats be presented:a) For example the hook-like structures on A-tubules have been observed in degenerating ciliary axonemes in adult *C. elegans*(for example: arl-13, nphp-4, ccpp-1), mouse (for example, Arl13b/hennin mutant), and zebrafish (Fleer mutant). Findings in this report suggest that these hook-like structures observed in other systems may represent a defect in development and not degeneration, and that cilia employ an evolutionarily conserved mechanism to construct the B-tubule. This would be a major change in how we view, interpret, and seek treatment options for ciliopathies.b) Another example, the authors observe vesicles within the *C. elegans* TZ and conclude "that the ciliary gate function of *C. elegans* TZs may be partly distinct from that in vertebrate cilia." An alternative possibility is that this study has provided unprecedented insight to multiple ciliated cell types and that this is not a worm oddity. Rather, it is technically impossible to do the same experiment in vertebrates (ie survey 10-12 vertebrate ciliated cell types at the same time in a developing embryo and in the adult). Because of the neuroanatomy of the worm (all the cilia are in the head) and studies by the Nicastro and Sengupta labs, we now have an appreciation of the diversity and complexity of developing and adult *C. elegans* cilia – there is no reason to think that vertebrate cilia are less complex. We just can't see it yet.

---

## [Author Response]

*The manuscript by Nechipurenko et al. is a logical follow up from authors' previous eLife paper defining an ultrastructural map of C. elegans sensory cilia and glia. Here, the authors again use ssTEM and ssET to capture early steps of ciliogenesis in rapidly forming cilia in C. elegans embryos, which is a difficult and heroic accomplishment in microscopy. The authors make a number of very interesting and unexpected findings that are worthy of publication pending some revisions.*

*1) Many of the inferred temporal events presented in the concluding model appear somewhat speculative in the absence of some type of complementary real-time visualization using light microscopy. The authors need to better emphasize the caveats associated with their deductions.*

We now emphasize this very valid caveat in the legend to Figure 2. We also note this caveat in the Discussion.

*2) Some of the implications of the work seem somewhat overstated – e.g. in the Discussion section the authors conclude that "[…]cilia in adult C. elegans […] possess a modified BB[…]that[…]provide(s) a site for docking, assembly and turnaround of IFT proteins…" but what is the evidence that these 3 processes require the modified BB? the authors should consider discussing their speculations about the import of the IFT machinery in the context of the paper by Prevo et al., 2015, Nat Cell Biol, 17, 1536 which reports the specific role of heterotrimeric kinesin-2 in importing the IFT machinery through structures described in the current paper at the base of the cilium.*

Our statement was based on work from the Dammermann and Hu labs. These groups recently reported that in animals mutant for the centriolar outer wall-associated protein HYLS-1, the docking and import of IFT machinery, and formation of the transition zone (TZ) and axoneme are severely impaired (1). HYLS-1 is also required for the formation of transition fiber (TF)-like structures comprised of the DYF-19/FBF1 protein among others. In an earlier report, the Hu lab showed that IFT particle entry into the cilium is also compromised in *dyf-19* mutants (2). Together, these papers suggest that at least some proteins localized to the modified BB region are important for distinct aspects of IFT. It is possible that the structural defects in *hyls-1* and *dyf-19* mutants compromise IFT particle entry by affecting kinesin-II-dependent transport of IFT trains through the ciliary base and TZ (3). We have now modified this and related sentences in the manuscript.

*3) Some implications of the authors findings are not emphasized in this manuscript and the authors are encouraged to discuss implications of their findings that will appeal to a broader audience. Overstatements should be avoided and caveats be presented:a) For example the hook-like structures on A-tubules have been observed in degenerating ciliary axonemes in adult C. elegans (for example: arl-13, nphp-4, ccpp-1), mouse (for example, Arl13b/hennin mutant), and zebrafish (Fleer mutant). Findings in this report suggest that these hook-like structures observed in other systems may represent a defect in development and not degeneration, and that cilia employ an evolutionarily conserved mechanism to construct the B-tubule. This would be a major change in how we view, interpret, and seek treatment options for ciliopathies.*

This is an intriguing suggestion. As noted in this manuscript, the inner AB junction is open initially forming A-tubules with the associated hooks originating at the outer AB junction. The inner junction closes later in development to form the AB doublets. Thus, B-tubules that are open at the inner AB junction such as in Arl13b *hnn* mutant mice (4) and *nphp-4* mutant worms (5) could potentially represent a developmental defect in B-tubule formation. However, *ccpp-1* tubulin deglutamylase mutants in *C. elegans* exhibit a range of B-tubule defects complicating interpretation (6), and the *fleer* mutant in zebrafish that exhibits reduced tubulin polyglutamylation appears to show defects in the outer AB junction (7). Compared to the inner AB junction, the outer junction has been reported to be more fragile and susceptible to breakage during biochemical experiments (8-10) suggesting that the hook-like structures observed in *fleer* mutants may arise due to weakening of the outer AB junction as a consequence of altered tubulin posttranslational modifications. We have now added speculation regarding the ultrastructural defects in *hnn* and *nphp-4* mutants to the text.

*b) Another example, the authors observe vesicles within the C. elegans TZ and conclude "that the ciliary gate function of C. elegans TZs may be partly distinct from that in vertebrate cilia." An alternative possibility is that this study has provided unprecedented insight to multiple ciliated cell types and that this is not a worm oddity. Rather, it is technically impossible to do the same experiment in vertebrates (ie survey 10-12 vertebrate ciliated cell types at the same time in a developing embryo and in the adult). Because of the neuroanatomy of the worm (all the cilia are in the head) and studies by the Nicastro and Sengupta labs, we now have an appreciation of the diversity and complexity of developing and adult C. elegans cilia – there is no reason to think that vertebrate cilia are less complex. We just can't see it yet.*

The reviewer makes a good point. Given that this was pure speculation and somewhat tangential to the main points of the manuscript, we deleted this section of the Discussion.

References

1. Wei Q, Zhang Y, Schouteden C, Zhang Y, Zhang Q, Dong J, et al. The hydrolethalus syndrome protein HYLS^-1^ regulates formation of the ciliary gate. Nat Commun. 2016;7: 12437.

2. Wei Q, Xu Q, Zhang Y, Li Y, Zhang Q, Hu Z, et al. Transition fibre protein FBF1 is required for the ciliary entry of assembled intraflagellar transport complexes. Nat Commun. 2013;4: 2750.

3. Prevo B, Mangeol P, Oswald F, Scholey JM, Peterman EJ. Functional differentiation of cooperating kinesin-2 motors orchestrates cargo import and transport in *C. elegans* cilia. Nat Cell Biol. 2015;17: 1536-1545.

4. Caspary T, Larkins CE, Anderson KV. The graded response to Sonic Hedgehog depends on cilia architecture. Dev Cell. 2007;12: 767-778.

5. Jauregui AR, Nguyen KC, Hall DH, Barr MM. The *Caenorhabditis elegans* nephrocystins act as global modifiers of cilium structure. J Cell Biol. 2008;180: 973-988.

6. O'Hagan R, Piasecki BP, Silva M, Phirke P, Nguyen KC, Hall DH, et al. The tubulin deglutamylase CCPP-1 regulates the function and stability of sensory cilia in *C. elegans*. Curr Biol. 2011;21: 1685-1694.

7. Pathak N, Obara T, Mangos S, Liu Y, Drummond IA. The zebrafish *fleer* gene encodes an essential regulator of cilia tubulin polyglutamylation. Mol Biol Cell. 2007;18: 4353-4364.

8. Witman GB, Carlson K, Berliner J, Rosenbaum JL. *Chlamydomonas* flagella. I. Isolation and electrophoretic analysis of microtubules, matrix, membranes, and mastigonemes. J Cell Biol. 1972;54: 507-539.

9. Stephens RE. Thermal fractionation of outer fiber doublet microtubules into A- and B-subfiber components. A- and B-tubulin. J Mol Biol. 1970;47: 353-363.

10. Linck RW, Langevin GL. Reassembly of flagellar B (α β) tubulin into singlet microtubules: consequences for cytoplasmic microtubule structure and assembly. J Cell Biol. 1981;89: 323-337.